# Types of Offers of Combustible Cigarettes, E-Cigarettes, and Betel Nut Experienced by Guam Youths

**DOI:** 10.3390/ijerph20196832

**Published:** 2023-09-26

**Authors:** Francis Dalisay, Scott K. Okamoto, Jane Teneza, Christina Dalton, Kayli Lizama, Pallav Pokhrel, Yoshito Kawabata

**Affiliations:** 1College of Liberal Arts and Social Sciences, University of Guam, Mangilao, GU 96923, USA; kawabatay@triton.uog.edu; 2Population Sciences in the Pacific Program (Cancer Prevention in the Pacific), University of Hawai’i Cancer Center, Oahu, HI 96813, USA; okamotos@hawaii.edu (S.K.O.); ppokhrel@cc.hawaii.edu (P.P.); 3Minority Health Research Training Program, Department of Tropical Medicine Medical Microbiology and Pharmacology, John A. Burns School of Medicine, University of Hawai‘i at Mānoa, Honolulu, HI 96813, USA; teneza20@hawaii.edu (J.T.); cdalton@muhlenberg.edu (C.D.); 4Pacific Island Partnership for Cancer Health Equity, University of Guam Cancer Research Center, Mangilao, GU 96913, USA; dornonk@triton.uog.edu

**Keywords:** adolescent, e-cigarette use, betel nut use, combustible cigarette use, prevention, offer scenarios, culturally specific

## Abstract

The present study examined types of scenarios in which Guam youths are offered tobacco—namely, combustible cigarettes and e-cigarettes—and betel (areca) nut. We conducted 10 focus groups with public middle school students (*n* = 34) from Guam. Results suggested that the types of offer scenarios of combustible cigarettes, e-cigarettes, and betel nut referenced by the students fall into two categories—direct-relational offers and indirect-contextual offers. The results also suggested that both categories of offer scenarios were more likely to occur in school rather than in other locations such as the home. Family members were more likely to make offers than other types of people. Indirect-contextual offers were more easily avoidable depending on the substance offered, the location where the offer took place, and the person making the offer. Based on the findings, we provide brief suggestions on developing a school-based prevention curriculum focused on training young adolescents from Guam on ways to resist offers of cigarettes, e-cigarettes, and betel nut.

## 1. Introduction

Guam, a U.S.-Affiliated Pacific Island (USAPI) located in the Western Pacific, has faced significant health disparities associated with the use of tobacco products and betel (areca) nut. Recent research indicates that Guam youths show higher rates of tobacco product use, including the use of combustible cigarettes and e-cigarettes, compared with their counterparts living in the continental U.S. [1]. Data from the most recent Guam State Epidemiological Profile [2] indicate that youths in Guam reported high rates of combustible cigarette and electronic cigarette (or vape) use. Particularly, the rates of cigarette smoking initiation before the age of 13 among Guam youths are more than twice the U.S. median; roughly 24% of high school and 13% of middle school students on the island report having smoked a cigarette before age 13. Additionally, the Guam Epidemiological Profile reveals that the prevalence of e-cigarette consumption is higher in Guam than the U.S. median, with about 14% of high school students in Guam and 10% of middle school students engaging in smokeless tobacco [2]. The same data indicate that around 66% of middle school students in Guam and about 36% of high school students have reported using e-cigarettes during their lifetime; about 27% of Guam high schoolers and roughly 24% of middle schoolers were identified as regular e-cigarette users. These proportions exceeded the U.S. median [2]. Guam’s youths also report consuming betel nut [1,3], which is the fruit of the *Areca catechu* palm containing an addictive chemical called arecoline. Betel nut has been identified as a Group 1 carcinogen by the World Health Organization [4]. The estimate of regular betel nut chewers among adults in Guam is about 11% [5], and Guam adolescents who chew betel nut start using it at around 11 years of age [3]. 

Research reveals adolescence is a critical period when youths from Guam and other USAPIs begin experimenting with addictive substances such as tobacco products, namely e-cigarettes, combustible cigarettes, and betel nut [1]. Prevention efforts should thus be aimed at Guam youths to mitigate the initial and continuous use of tobacco and betel nut. One promising approach is to develop a culturally grounded youth substance use prevention program for Guam youths, as prior research has found such programs to be effective in reducing youth substance use or its risk factors [6,7]. However, there is currently a dearth of tobacco and betel nut prevention programs specifically designed and tailored for the cultures of young adolescents from the Pacific. This is primarily attributable to the limited understanding of the culturally specific scenarios in which Guam youths are offered tobacco and betel nut. Thus, there is a critical need to gain a wider breadth of knowledge of such scenarios to design and develop culturally responsive programs aimed at preventing e-cigarette, cigarette, and betel nut use. 

The purpose of the present study is to examine the scenarios in which youths from Guam encounter or are offered e-cigarettes, combustible cigarettes, and betel nut. This study investigates the type of offers of combustible cigarettes (henceforth, cigarettes), e-cigarettes, and betel nut experienced by early adolescents in Guam, the locations where such offers are commonly made, and the typical persons who make such offers. The present study is part of a larger formative research project intended to inform the development of a culturally responsive prevention program focused on training young adolescents from Guam on ways to navigate through scenarios of substance use offers and to resist substances.

### Direct-Relational Offers and Indirect-Contextual Offers 

As we noted above, there is a lack of understanding about scenarios in which Guam youths are offered tobacco and betel nut. However, prior research involving Native Hawaiian youths, conducted by Helm et al. [8], provides a framework for examining drug offer scenarios experienced by USAPI youths. Their research, which employed focus groups conducted in Hawai‘i, revealed that the bulk of drug offer scenarios experienced by Native Hawaiian youths can be categorized as either direct-relational or indirect-contextual offers. The researchers explained that these two types of offers should be distinguished from actively seeking out drugs, which involves voluntarily seeking out substances. Helm et al.’s [8] research also suggests Native Hawaiian youths are more likely to encounter drugs through offers rather than through actively seeking them. Indeed, among the 47 youths who participated in their study, only one participant reported actively seeking out drugs. 

With the above framework as a guide for types of offers that USAPI youths may likely experience, the present study focuses on examining direct-relational and indirect-contextual offers experienced by Guam youths. Direct-relational (DR) offers occur when the substance offeror, who has an established relationship with the participant, explicitly offers the substance to the participant. In this type of offer, the substance offeror directly interacts with the participant, making an explicit invitation to use drugs. This type of offer generally occurs in school or home environments and is made by classmates, friends, cousins, uncles, parents, and grandparents [8]. For example, Helm et al.’s [8] study documented the experience of an 8th-grade girl who, during recess, was explicitly offered an alcoholic drink by her friends. 

Indirect-contextual (IC) offers occur when no explicit offer of a drug is made, but instead, participants find themselves in a situation or in an environment where drugs are involved, and there is an implicit demand for youths to use them. IC offers tend to occur in settings such as parties, school bathrooms, or areas behind schools [8]. The narratives reported by middle school youths in Helm et al.’s [8] study suggest that when it comes to IC offers, there is a certain social and cultural expectation involving drug use. Similarly, USAPI youths may encounter situations (knowingly or unknowingly) where drug use will occur, and there may be an underlying pressure, possibly stemming from social norms and peer pressure, to experiment with drugs, even without an explicit offer to use drugs. 

According to Helm et al.’s [8] research, there are two types of indirect-contextual offers: avoidable and unavoidable offers. Avoidable IC offers refer simply to scenarios that a person can avoid. An example of an avoidable IC offer would be a youth deciding to decline an invitation to attend an event if they knew that others who would also be attending the event would likely be smoking and drinking. On the other hand, unavoidable IC offers typically involve being around family or adult relatives at social events where leaving the context is not an option [8]. The youth is required to be in a particular place, usually at school or with family members at home. Examples of unavoidable offers could include parents smoking at home or being around older relatives using substances at family gatherings. These situations are considered unavoidable because youths have to stay in a particular setting. 

The present study also investigates the locations where offers of e-cigarettes, cigarettes, and betel nut are likely to occur for Guam youths and the persons who are likely to offer these substances. First, regarding locations, youths spend most of their day at school and at home. Thus, common locations where substance use offers may occur include schools (e.g., during a free period, recess, or after school activity) and at a home setting (e.g., in one’s own house or at a friend’s house) [8]. In a study of Guam adolescents, Pokhrel et al. [1] found that ease of access to cigarettes at home and at school was positively associated with the likelihood of using cigarettes in the past month. A similar positive association was found between ease of access to betel nut at home and school and the likelihood of using betel nut. In another study, Dalisay et al. [3] conducted interviews with 20 adolescents who were current betel nut users. Dalisay et al. [3] found that some of the adolescents were first introduced to betel nut at their schools through offers from friends, while others were first introduced to betel nut at home, where the substance was easily accessible (e.g., left on top of a table). 

Second, peers exert a strong influence on youths’ behaviors. Youths who have friends who use drugs are also more likely to engage in drug-using behavior themselves [9]. Helm et al. [8] found that Native Hawaiian youths were typically offered drugs by their peers, such as close friends, classmates, and other students in school. Similarly, Pokhrel et al. [1] found that adolescents in Guam who had peers who used cigarettes and betel nut were also more likely to have used cigarettes and betel nut themselves in the past month. Also, Guam adolescents who had easier access to both cigarettes and betel nut through their friends were more likely to have used cigarettes and betel nut in the past month [1]. 

In addition to peers, family members could shape youths’ perspectives toward drugs. Parenting styles, the smoking habits of parents, and family structure have been shown to influence smoking and drinking in adolescents [10,11]. Among early adolescents in Guam, family members’ use of cigarettes and betel nut, having parents who permit the use of the substances, and ease of access to these substances through family members are positively associated with the use of these substances in the past month [1]. 

In sum, the above literature suggests that the types of offer scenarios experienced by youths in Guam could be categorized as direct-relational or indirect-contextual offers. The present study investigates the following research questions (RQs) regarding cigarette, e-cigarette, and betel nut offer scenarios experienced by Guam adolescents:

RQ1: To what extent do Guam adolescents experience direct-relational (DR) and indirect-contextual (IC) offers? 

RQ2: Who are the typical types of people making such offers? 

RQ3: Where do such offers commonly take place? 

## 2. Materials and Methods

### 2.1. Procedure and Participants

Data were collected between the months of February 2021 to April 2021. University of Guam’s Institutional Review Board, which serves as the regional IRB, approved this study before research staff began recruiting participants. Students in the Guam Department of Education (GDOE) were invited to participate in this study. School teachers assisted in recruiting student participants by providing information to their classes. Using convenience sampling, students interested in participating in this study were recruited after obtaining written parental consent and student assent. These forms provided information about confidentiality and participants’ rights. The following were the eligibility criteria: (a) the participants needed to be between 11 and 14 years of age and (b) currently enrolled in a GDOE middle school. The final sample included 34 students who were between 11 and 14 years old and who were from five different public middle schools. Participants were given $25 gift certificates to a local store upon completion of the study. 

Ten semi-structured focus group interviews were conducted. Groups were separated by gender, girls (*n* = 20) and boys (*n* = 14), with an average of four participants in each group. Due to the focus on tobacco product use for the study, youths were separated into gender-specific focus groups. This was done so that youths were more comfortable in discussing sensitive topics, such as tobacco and drug offers in the school. Indeed, research suggests that qualitative researchers may be able to obtain a greater diversity in opinions from adolescents by the use of single-gender focus groups [12].

Group sizes were intentionally small due to the sensitive nature of the topic (substance use) and the need to promote and establish trust between the research staff and youth participants. Of the overall participants, 12% were in grade 6, 56% in grade 7, and 29% in grade 8. Ethnically, participants identified themselves as CHamoru (32%), Filipino (29%), Chuukese (21%), Pohnpeian (6%), Palauan (3%), and mixed ethnicity (9%). 

Due to COVID-19-related government restrictions during the time of data collection, in-person focus group discussions were not possible. Discussions were instead held through Zoom, an online teleconferencing platform. Laptops and portable Wi-Fi devices were provided to participants who did not have access to laptops and/or internet at home. 

For each group of boys and girls, trained adult males and females moderated the discussions, respectively. Each focus group discussion was approximately 60 min long. Moderators asked the following questions and appropriate probes when necessary:Have you or kids that you have known ever been offered e-cigarettes, tobacco, or betel nut?If you were offered e-cigarettes, tobacco, or betel nut, what did you do?If you know of other kids that were offered these substances, what did they do?Have you or kids that you have known ever been invited to go with other kids who planned to use e-cigarettes, tobacco, or betel nut?If you were invited, what did you do? If kids you have known were invited, what did they do?Is it hard to resist e-cigarettes, cigarettes or other tobacco products, and betel nut? Why or why not?

During the interviews, participants were asked to choose an alias that they could be addressed with and were given the option to keep their cameras off. These steps were taken to promote confidentiality and to control for social desirability effects. Additionally, the focus group questions were drafted and facilitated to ensure a non-biased conversation with the participants regarding e-cigarette or betel nut use. 

### 2.2. Data Analysis

The research team used the Zoom platform to record and transcribe all 10 of the focus groups. Initial transcriptions were extracted from Zoom and were reviewed and edited by at least two members of the research team for data accuracy. Data were analyzed using NVivo 12. To analyze the data, we employed the coding framework established by Helm et al. [8] in their study of scenarios in which Hawai‘i adolescents were offered drugs. We first coded the scenario as direct-relational or indirect-contextual using Helm et al.’s [8] framework. Using this same framework, indirect-contextual offer scenarios were also coded as either avoidable (i.e., the youth was not required to be at the place where the IC offer was made and, thus, leaving the context was an option; e.g., the youth simply declined the offer, walked away, or left) or unavoidable (i.e., the youth was required to be there and, thus, leaving the place was not an option; the youth appeared to be stuck at the place). Next, for each scenario, the research team coded for (a) the type of substance being offered (i.e., e-cigarettes, combustible cigarettes, and betel nut), (b) the locations where the offer occurred (e.g., home, school, or other location), and (c) the person making the offer (i.e., family member, peer-friend, peer-other student at school, or other persons who were neither family nor friend). The research team established intercoder reliability and validity by collectively coding the first two transcripts to identify and define all the codes. Each of the remaining transcripts was then coded by at least two trained research assistants. The lead researcher (first author) then went through all coded transcripts to verify the validity of the coded references. All non-identically coded references were identified and discussed (i.e., added or omitted in the data set) by members of the team until consensus was reached. 

## 3. Results

The present study’s analysis revealed that 100% (*n* = 10) of the focus groups had referenced being offered at least one of the three substances—cigarettes, e-cigarettes, and betel nut. As shown in Table 1, 90% of the focus groups referenced direct-relational (DR) offer scenarios, and 100% referenced indirect-contextual (IC) offer scenarios. Among the locations where offers were likely to occur, school was the most frequently referenced, with 90% of the focus groups mentioning this location. Furthermore, with respect to the persons making offers, the findings suggest that both types of offer scenarios were most likely to be made or occur among or with family members. Below, we report results for each of the three categories of offers, and in line with our RQs, we report specific findings for the type of substances being offered, locations where offers are typically made, and the persons making the offers. 

### 3.1. Direct-Relational Offers

#### 3.1.1. Substance 

As shown in Table 2, e-cigarettes were the most frequently referenced substance that was being offered through direct-relational offers (with 90% of the focus groups referencing a direct-relational offer of e-cigarettes). On the other hand, as shown in Table 2, 50% of the focus groups reported experiencing DR offers of cigarettes, and 50% reported DR offers of betel nut. 

#### 3.1.2. Location

A majority of DR offers occurred at school (referenced by 60% of the focus groups), in specific locations such as a school restroom, a basketball court, and at the back of a classroom during class. In addition, the results revealed that 10% of the focus groups referenced home as a location of DR offers. Yet, 40% of the focus groups made references to “other locations” other than school or home, where DR offers are likely to occur. These other locations include around the neighborhood, a basketball court at a community center, and inside a car. One female participant mentioned she witnessed a DR offer occurring over a social media site. 

#### 3.1.3. Persons

Not surprisingly, the results indicate that the majority of direct-relational offers were made by family members (referenced by 80% of the focus groups), followed by peers, i.e., close friends (referenced by 60% of the focus groups) and other students (referenced by 60% of the focus groups); none (0%) of the focus groups referenced that people other than family members or peers made DR offers. Family members who were referenced as making DR offers tended to be older relatives, such as cousins, uncles, and aunts. One male participant said that he was around 10 or 12 years old when his uncles offered him an e-cigarette at a family party. Another participant, a female, reported hearing a story from a friend whose father had made a DR offer of betel nut:

Participant: Well, I had a friend who told me a story.

Facilitator: Hmm, what’s the story?

Participant: Um, she told me that, um, her dad—her dad, um, gave her to try and she didn’t take it and she told me—when the next day she came, she told me that her dad got really mad at her because she didn’t take it, and I just felt so bad for her.

The example directly above suggests that declining a betel nut offer from an adult may be unacceptable for some youths in Guam. The above example also suggests that some parents in Guam may be permissive regarding their child’s use of substances. Indeed, a few other participants explained that some youths may have parents who condone substance use behaviors. For example, a female participant stated,

Hmm, I think that some of the kids, like their parents don’t really care that much, um, about what they do and stuff, and so they would just buy whatever the kids want, just so that way the kids don’t bother them and stuff, and so the kid would just get it from their parents and their parents would actually know.

On the other hand, peers were also a source of DR offers. For instance, a female participant reported that she knew of others her age who currently used e-cigarettes and cigarettes and that they had older friends who purchased these products for them: 

Uh, I know a lot of people who have like older friends that are probably like in high school and then their high school friends have like people who are like seniors or like above 18 that can purchase it and they give it to the high schoolers and then the high schoolers would probably like hand it or like, you know, lend it to them.

Yet, a few participants reported that DR offers were made by peers other than close friends. For example, one male participant explained that he had experienced an offer of e-cigarettes at school during lunchtime from “other kids” whom he did not identify as being his close friends. The participant explained,

Participant: It was, it was um. It was during like lunchtime and like during our break I was playing basketball and then I hurt my foot so I had to sit down for a bit and then that’s when I was offered [to use an e-cigarette].

Facilitator: When you sat down were there other kids that were um sitting around the basketball courts watching kids play is that what happened?

Participant: Yeah there was a couple of them there. They were just chilling next to the uh basketball hoop.

### 3.2. Indirect-Contextual Offers

#### 3.2.1. Substance 

As shown in Table 2, all focus groups (100%) referenced experiencing IC offers of cigarettes and e-cigarettes, while 80% of the focus groups referenced experiencing IC offers of betel nut. Also shown in Table 2, a greater number of focus groups referenced IC offers of e-cigarettes as being avoidable (80%) than unavoidable (60%). Moreover, as Table 2 shows, although more focus groups referenced IC offers for cigarettes as being unavoidable (80%) rather than avoidable (70%), there was only a small difference (10%) between the number of focus groups referencing unavoidable and the number of focus groups referencing avoidable offers of cigarettes. Finally, as Table 2 shows, there were no differences in the numbers of focus groups that referenced IC offers for betel nut to be either avoidable or unavoidable, as both types of IC offers were referenced by 70% of the focus groups, respectively. 

#### 3.2.2. Location 

The results shown in Table 3 indicate that the top three locations where indirect-contextual offers were likely to occur included, in descending order, school (referenced by 90% of the focus groups), at home (referenced by 80% of the focus groups), and other locations (referenced by 70% of the focus groups). Typical “school” locations where IC offers occurred included bathrooms, gyms, and areas within the school where adults such as teachers and teacher aides were not present. IC offers at “home” tended to occur outdoors (e.g., in a gazebo located outside a home or at family parties). The “other” non-school/non-home locations where IC offers were reported to occur included convenience stores, the local mall, and abandoned buildings. As shown in Table 3, more focus groups referenced that IC offers occurring at school were avoidable (90% of the focus groups) rather than unavoidable (30% of the focus groups). On the other hand, more focus groups referenced that IC offers occurring at home and other locations were unavoidable (50% of focus groups) rather than avoidable (20% of focus groups). Also, while there were more focus groups who referenced IC offers made in locations other than school or home as being avoidable (40% of focus groups) rather than unavoidable (30% of focus groups), the difference in the number of focus groups was small (10%). By comparison, there was a 60% difference between the percentage of focus groups that referred to IC offers being avoidable and unavoidable in the school setting. Avoidable IC offers at school tended to occur in contexts where the participant could have easily walked away or simply declined to go in the first place. For example, one male participant, a 6th grader, mentioned that he had once witnessed a group of 7th and 8th graders vaping inside the bathroom at his school. Yet, the participant said he simply walked away from the context as he explained that he had just finished lunch and was heading to his next class. Another participant, a female, said she was with her friends at school, and while they were standing beside a locker room at their school’s gym, she smelled someone smoking a cigarette nearby: 

Facilitator: What did you do when you smelled that [the cigarette smoke]?

Participant: I just went around and told my friends, like I was just asking, “Do you smell that? Do you smell that?” And I was just like—I just covered my nose and just try to stay away from that place because like it would, it would really bother me. That’s why.

On the other hand, unavoidable IC offers tended to occur at home with family members present. For example, a female participant commented whenever she went to her grandmother’s ranch house, she would always witness people chewing betel nut in the kitchen: 

Facilitator: Oh okay. Okay. So, do you ask them?

Participant: Oh yeah, I always ask them and they’re like, “Oh, this is pugua, pugua” [pugua is the CHamoru name for betel nut. CHamorus are the native inhabitants of Guam.]. And ‘cause we actually have a ranch where there’s a lot of pugua in there and they, they’re, they’ll, they’re always gonna go to that ranch and they will take like pugua and other things. And ‘cause I’ve been living here in Guam for like 2 years, so like every time I go to the house [i.e., her grandmother’s ranch house], um, I always see them and I’ve always been curious about what it tastes and they’re like [shrugs] “hmm.” They’re like, yeah. While they’re cooking, they’re chewing.

#### 3.2.3. Persons

As shown in Table 4, most of the IC offers occurred among or with family members or among other students in school. Specifically, 100% of the focus groups referenced that IC offers were made among or with family members, and 80% referenced that IC offers were made by other students at school. IC offers were less likely to come from close friends (referenced by 40% of the focus groups) and even more less likely to come from other non-family members and non-friends (referenced by 10% of the focus groups). Furthermore, the results in Table 4 show that a greater number of focus groups referenced IC offers from peers (close friends and other students) as more easily avoidable (40% for close friends; 80% for other students) than unavoidable (10% for close friends; 20% for other students). On the other hand, more focus groups referenced IC offers from family members as being unavoidable (referenced by 80%) rather than avoidable (referenced by 30% of the focus groups). These results seem to reinforce Helm et al.’s [7] findings that avoidable IC offers tend to occur among peers, whereas unavoidable IC offers tend to occur among family. For example, a male participant was asked whether he felt peers who made offers to him might retaliate if he avoided or turned down an IC offer: 

Facilitator: Have you ever felt threatened that they [his peers] might sort of retaliate against you ‘cause you saw them or is it, that doesn’t even, you don’t even think about that?

Participant: No ‘cause I try to ignore… I try to avoid those types of people anyway.

This situation suggests that declining an offer from a peer can be straightforward. On the other hand, unavoidable IC offers made by family tended to involve situations where the youth was required to stay put. An example of an unavoidable IC offer was provided by a male participant, who explained that during sleepovers at his cousin’s house, he would witness his cousin’s friends vaping. In this example, it was not possible for the youth to leave the sleepover, and thus, this IC offer could not be avoided: 

Participant: Usually when I go up to Dededo [a northern village in Guam], um, and I sleepover at my cousin’s house, uh, sometimes they like to invite some of their friends and I see like vapes sometimes, and I see them outside vaping.

## 4. Discussion

The present study examined typical scenarios in which Guam youths are offered tobacco products—namely, e-cigarettes and combustible cigarettes—and betel nut. Our study investigated the types of scenarios in which Guam adolescents are offered e-cigarettes, cigarettes, and betel nut, where such offers typically occur, and the people likely to make such offers. We conducted 10 focus groups comprising early adolescents in Guam. Our study provided various insights into substance offers experienced by Guam youths that can be summarized into five key findings. Below, we discuss these findings, and based on these findings, we provide brief suggestions relevant to the development of a culturally responsive substance use prevention program for Guam youths. 

First, the present study indicated that Guam youths encounter cigarettes, e-cigarettes, and betel nut through direct-relational (DR) and indirect-contextual (IC) offers. These results reinforce Helm et al.’s [8] previous findings, which also showed that DR and IC offers made up the bulk of types of offers referenced by Native Hawaiian youths. They also suggest that Guam adolescents need exposure to substance use prevention to deal with DR and IC offers. This can be achieved through the development of a culturally grounded substance use prevention curriculum for Guam youth, which could employ the use of realistic, culturally relevant videos, discussions, and activities to promote effective drug resistance strategies for these youth. Such strategies, which include refusing or saying no, leaving and avoiding situations where offers are made, or providing an explanation to turn down an offer, have been found to be effective in reducing substance use among youths in both the U.S. and abroad [6,13,14,15]. 

Second, the present study revealed that with respect to location, “school” was the most frequently referenced context where offers occurred. School is the location where youths in Guam may be more likely to be introduced to cigarettes, e-cigarettes, and betel nut. This finding suggests the relative priority of resistance skills training in the school context for tobacco and betel nut prevention for Guam adolescents. 

Third, the study revealed that substance use offers are most likely to come from family members. On balance, our results suggest that offers from family members were not easy to avoid. These findings extend previous research by Pokhrel et al. [1], which has found that Guam youths who had easier access to cigarettes and betel nut through family and friends were also more likely to have recently used cigarettes and betel nut. Yet, we should note that IC offers were unavoidable for family members but more avoidable for peers. However, considering Guam’s predominately collectivistic culture [16,17], youths in Guam could possibly be more susceptible to norms from their family and close friends, who exert considerable peer pressure on youths and cultural norms, rather than the overall perceived prevalence of use of a given substance among their peers in school. Finally, although IC offers were more frequently reported to come from family members, IC offers at home were reported to occur less frequently than IC offers made at school. Broadly speaking, it is quite common for students in Guam to have older cousins or siblings attending the same schools as them. Thus, offers in school could have also come from these older cousins or siblings. These cousins or siblings may have been less willing to experiment with substances at home due to the presence of adult family members. They may have been more willing to experiment with substances in places outside of the home, such as school.

These findings suggest the need for family-based tobacco and betel nut prevention for Guam youth. This type of intervention might target parents or close adult extended family members and the impact that family norms and permissiveness around tobacco use may have on Guam youths’ decisions to use substances. On the other hand, due to the cultural and traditional practice of betel nut chewing in the Pacific region [5,18,19,20], there is a potential that betel nut use may be viewed by Guam youths as more acceptable than tobacco use, and thus, betel nut use behaviors may be more difficult to change. We recommended that more research be conducted to examine in-depth, both among Guam adolescents and their parents’ awareness of the risk of betel nut chewing. This can further inform the development of interventions aimed at reducing or preventing adolescent betel nut use. Such interventions could be informed by existing programs aimed at reducing rates of betel nut use among adults [21]. 

Fourth, our study suggests youths may be more likely to indirectly encounter substance use offers. Given the prevalence of IC offer types, future research can prioritize these offer types to develop more effective prevention strategies for addressing youth substance abuse. For instance, indirect-contextual offers can be addressed through training youths in using outside-avoidance skills. These skills are those in which youths can predict future problem situations (e.g., a family party where drug use might occur) and avoid these situations before they occur (i.e., from outside the situation). 

Fifth, our findings suggest that indirect-contextual offers from peers (i.e., close friends and other students) are more easily avoidable, while IC offers from family members are more difficult to avoid. These results seem to reinforce Helm et al.’s [8] findings that avoidable IC offers tend to occur among peers, whereas unavoidable IC offers tend to occur among family members. These types of offers can be addressed by training youths in inside-avoidance skills. These are skills that youths use to avoid substance use when they are already “inside” the situation and can range from making an excuse to leaving the situation. Thus, a similar approach could be adapted for youth substance youth prevention programs in Guam. 

Our study generated promising results and had two primary strengths. First, to our knowledge, our study is the first to have reported the types of substance offer scenarios typically experienced by Guam adolescents. Second, the qualitative nature of our study was a strength that allowed us to examine in greater detail the types of substances (tobacco, betel nut) that Guam adolescents are offered, where such offers are typically made, and who are making such offers. Taken together, our formative findings could be useful for further prevention research and the development of policies concerning adolescents in Guam and the USAPI more broadly.

Nevertheless, some limitations should be acknowledged. First, given that the target populations of the study were adolescents in Guam, the findings could not be generalized and applied to other age and cultural groups. Second, our study focused specifically on offers of tobacco products and betel nut. The extent to which our findings can be applied to other types of substances, such as alcohol and marijuana, should be examined by future research. Third, while our study was able to identify the persons making offers, unfortunately, we did not probe for the types of key persons who may be persuading the offeror not to offer the products to adolescents or what would inspire the key persons who may be persuading the offeror not to offer these products to adolescents. We recommend future research to examine these key persons, as such information could be helpful in developing prevention campaigns. 

Finally, we had issues with using a virtual format to conduct our focus groups. These ranged from issues with data management (e.g., linking the recorded video content with youth responses within the chat in Zoom), distractions from other family members during the focus groups, and internet connectivity issues. 

## 5. Conclusions

In sum, the present data indicated offer scenarios of combustible cigarettes, e-cigarettes, and betel nut for Guam youths could occur as either direct-relational (explicit) offers or indirect-contextual (implicit) offers. Such offers are more likely to occur in school rather than in other locations, such as the home, even though the majority of offers appear to be made by family members (e.g., cousins, siblings). Tobacco and betel nut prevention programs designed for Guam youths may benefit from focusing on training youths on strategies to resist direct-relational and indirect-contextual offers in culturally appropriate ways.

## Figures and Tables

**Table 1 ijerph-20-06832-t001:** Percentages of focus groups referencing direct-relational offers and indirect-contextual offers.

Direct-Relational	Indirect-Contextual	Avoidable Indirect- Contextual	Unavoidable Indirect- Contextual
90%	100%	90%	90%

**Table 2 ijerph-20-06832-t002:** Percentages of focus groups referencing direct-relational offers and indirect-contextual offers according to type of substance.

	Combustible Cigarettes	E-Cigarettes	Betel Nut
Direct-relational	50%	90%	50%
Indirect-contextual	100%	100%	80%
Avoidable Indirect-contextual	70%	80%	70%
Unavoidable Indirect-contextual	80%	60%	70%

**Table 3 ijerph-20-06832-t003:** Percentages of focus groups referencing direct-relational offers and indirect-contextual offers according to location.

	School	Home	Other Locations
Direct-relational	60%	10%	40%
Indirect-contextual	90%	80%	70%
Avoidable Indirect-contextual	90%	20%	40%
Unavoidable Indirect-contextual	30%	50%	30%

**Table 4 ijerph-20-06832-t004:** Percentages of focus groups referencing direct-relational offers and indirect-contextual offers according to person making the offer.

	Family	Peers-Close Friends	Peers-Other Students in School	Other-Non-Family, Non-Peers
Direct-relational	80%	60%	60%	0%
Indirect-contextual	100%	40%	80%	10%
Avoidable Indirect-contextual	30%	40%	80%	10%
Unavoidable Indirect-contextual	80%	10%	20%	0%

## Data Availability

The University of Guam Institutional Review Board does not permit the sharing of this study’s data due to confidentiality concerns.

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
