# Peer review of "Types of Offers of Combustible Cigarettes, E-Cigarettes, and Betel Nut Experienced by Guam Youths"

_ijerph, 2023, doi:10.3390/ijerph20196832_

Round 1

Reviewer 1 Report

Comments attached.

Consider reviewing the quality of English

Author Response

We thank Reviewer 1 for the invaluable comments and suggestions.

We much appreciate the time spent by Reviewer 1 in reading and reviewing our manuscript. 

Below, we have provided the comments of the Reviewer 1 followed by our responses. 

--

Reviewer 1:

  1. Introduction:

  1. Kindly mention the need explicitly.

Response: Thanks for this suggestion. On page 2, lines 60-62, we have revised the following sentence so that the need is explicitly stated: “Thus, there is a critical need to gain a wider breadth of knowledge of such scenarios to design and develop culturally responsive programs aimed at preventing e-cigarette, cigarette, and betel nut use.”

  1. Materials and methods (line number 160):

  1. Procedure and sample (line 160): Kindly elaborate the eligibility criteria during sampling. Whether tobacco users were recruited, etc.

Response: We added the following statement (see line 160) to specify the eligibility criteria: “The following was the eligibility criteria: a) the participants needed to be between 11 to 14 years of age and b) currently enrolled in a GDOE middle school.” 

  1. On what basis the genderwise grouping was done. Give reasons or reference.

Response: We have added the following in order to provide the basis for the use of genderwise grouping (please see lines 167-171):

Due to the focus on tobacco product use for the study, youth were separated into gender-specific focus groups. This was done so that youth were more comfortable in discussing sensitive topics, such as tobacco and drug offers in the school. Indeed, research suggest that qualitative researchers may be able to obtain a greater diversity in opinions from adolescents by the use of single-gender focus groups [11].

Reference:

  1. Daley A. M. Adolescent-friendly remedies for the challenges of focus group research. West J Nurs Res 2013, 35, 1043–1059.

  1. Kindly mention the type of study in the methodology and questionnaire if it is structured or semi structured or unstructured.

Response: We employed semi-structured interviews. We now specify this methodology on line 165: “Ten semi-structured focus group interviews were conducted.” 

  1. Mention the duration of the study.

Response: Regarding duration of the study, we now mention on line 154 that data were collected between the months of February 2021 to April 2021. We also mention on lines 182-183 that each focus group discussion was 60 minutes long.

III. Results:

  1. Line number 239-246:

Mention the table number.

Response: We have now added the Table number.

  1. Line number 292-300:

Mention the table number.

Response: We have now added the Table number.

  1. Discussion:

  1. Line number 436: Brief strengths of the study.

Response. We have added the following in lines 445-452: Our study generated promising results and had two primary strengths. First, to our knowledge, our study is the first to have reported the types of substance offer scenarios typically experienced by Guam adolescents. Second, the qualitative nature of our study was a strength that allowed us to examine in greater detail, the types of substances (tobacco, betel nut) that Guam adolescents are offered, where such offers are typically made, and who are making such offers. Taken together, our formative findings could be useful for further prevention research and the development of policies concerning adolescents in Guam and the USAPI more broadly.

Reviewer 2 Report

Smoking is important risk for many chronic diseases and cancers.  The higher smoking rates of Guam youth should be highly concerned. It's important to learn about the scenarios for the youth to get offers of cigarrettes and e-cigarrettes.  It would be more important to know if there is any solution to change the unavoidable IC offers to be avoidable. The following information should be collected and analyzed also:

1. Whose offers were not easy to avoid? Who are the key person to persuade the offeror not to  offer these products to adolescents?  What would inspire the offeror to give up such offers to adolescents?

2. If the IC offers were more common from other family members, why IC offers were not common at home? Is there any key care-givers of adolescents who can forbid such kind of offers at home?

3. Although betel nut is group A carcinogen, it's quite different from smoking. It maybe more related to local habits. The attitude/ risk awareness of betel nut might be rooted in local residents minds. It's another risk behavior to be intervened. It might be more difficult to change. I suggest to collect information about the risk awareness of betel nut both among Guam adolescents and their parents.

4. What are the common successful refusing reasons or useful ways for the adolescents to avoid such kind of smoking offers?

5. Why "training young adolescents from Guam on ways to navigate through substance use offers" other than training adolescents to avoid cigarrettes or e-cigarrettes smoking offer?

In addition, please check the proportion of unavoidable and avoidable IC offers. Are they all equal to 90%?

The English is good, only  few sentences are not easily understood.

Author Response

We thank Reviewer 2 for the invaluable comments and suggestions.

We much appreciate the time spent by Reviewer 2 in reading and reviewing our manuscript. 

Below, we have provided the comments of the Reviewer 2 followed by our responses. 

Reviewer 2:

Smoking is important risk for many chronic diseases and cancers.  The higher smoking rates of Guam youth should be highly concerned. It's important to learn about the scenarios for the youth to get offers of cigarrettes and e-cigarrettes.  It would be more important to know if there is any solution to change the unavoidable IC offers to be avoidable. The following information should be collected and analyzed also:

  1. Whose offers were not easy to avoid? Who are the key person to persuade the offeror not to offer these products to adolescents?  What would inspire the offeror to give up such offers to adolescents?

Response: We offer the following responses to each of the above separate questions.

  1. Whose offers were not easy to avoid?

On balance, our results suggest that offers from family were not easy to avoid. We have added this statement to our manuscript—see lines 416-417.

b and c. Who are the key person to persuade the offeror not to offer these products to adolescents? What would inspire the offeror to give up such offers to adolescents?

Unfortunately, our study did not probe the key persons who may be persuading the offeror not to offer these products to adolescents, or what would inspire the key persons who may be persuading the offeror not to offer these products to adolescents. We have added this as a limitation and have recommended further research in these areas. We have added the following (see lines 475-480):

“Third, while our study was able to identify the persons making offers, unfortunately, we did not probe for the types of key persons who may be persuading the offeror not to offer the products to adolescents, or what would inspire the key persons who may be persuading the offeror not to offer these products to adolescents. We recommend future research to examine these key persons, as such information could be helpful in developing prevention campaigns.”

  1. If the IC offers were more common from other family members, why IC offers were not common at home? Is there any key care-givers of adolescents who can forbid such kind of offers at home?

Response: Thanks for these questions. First, relatively speaking, IC offers were also common at home, but as we reported on our manuscript, IC offers were most likely to occur at school versus the home or other locations. According to our findings, 80% of the focus groups reported experiencing IC offers being made at home, whereas 90% of the focus groups reported experiencing IC offers being made at school, and 70% of the focus groups reported experiencing IC offers being made at other locations other than school or home. Second, broadly speaking, it is quite common for students in Guam to have older cousins or siblings attending the same schools as them. Thus, offers in school could have also come from these older cousins or siblings. As implied by Reviewer 2, these cousins or siblings may have been less willing to experiment with substances at home due to the presence of adult family members. They may have been more willing to experiment with substances in places outside of the home, such as school. We have added the following (lines 424-431):

Finally, although IC offers were more frequently reported to come from family members, IC offers at home were reported to occur less frequently than IC offers made at school. Broadly speaking, it is quite common for students in Guam to have older cousins or siblings attending the same schools as them. Thus, offers in school could have also come from these older cousins or siblings. These cousins or siblings may have been less willing to experiment with substances at home due to the presence of adult family members. They may have been more willing to experiment with substances in places outside of the home, such as school.  

  1. Although betel nut is group A carcinogen, it's quite different from smoking. It maybe more related to local habits. The attitude/ risk awareness of betel nut might be rooted in local residents minds. It's another risk behavior to be intervened. It might be more difficult to change. I suggest to collect information about the risk awareness of betel nut both among Guam adolescents and their parents.

Response: Thanks for this suggestion. On lines 439-446, we have included a statement with appropriate citations that due to the cultural and traditional practice of betel nut chewing in the Pacific region [5], there is a potential that betel nut use may be more acceptable than tobacco use, and thus, betel nut use behaviors may be more difficult to change. We recommended that more research be conducted to examine in depth, both among Guam adolescents and their parents’ awareness of the risk of betel nut chewing. This can further inform the development of interventions aimed at reducing or preventing adolescent betel nut use.

Reference

  1. Paulino, Y. C.; Hurwitz, E. L.; Ogo, J. C.; Paulino, T. C.; Yamanaka, A. B.; Novotny, R.; Wilkens, L. R.; Miller, M. J.; Palafox, N. A. Epidemiology of areca (betel) nut use in the Mariana Islands: Findings from the University of Guam/University of Hawai`i cancer center partnership program. Cancer Epidemiol 2017, 50(Pt

B), 241-246 doi: 10.1016/j.canep.2017.08.006

  1. Sotto, P. P.; Mendez, A. J.; Herzog, T. A.; Cruz, C.; Chennaux, J. S. N.; Legdesog, C.; Paulino, Y. Barriers to quitting areca nut consumption and joining a cessation program as perceived by chewer and nonchewer populations in Guam. Subst Use Misuse 2020, 55, 947-953. doi: 10.3380/fcomm.2022.960093

  1. Murphy, K. L.; Herzog, T. A. Sociocultural factors that affect chewing behaviors among betel nut chewers and ex-chewers in Guam. Hawaii J Med Public Health 2015, 74, 406-411.

  1. Narayanan, A. M.; Yogesh, A.; Chang, M. P.; Finergersh, A.; Orosco, R. K.; Moss, W. J. A survey of areca (betel) nut use and oral cancer in the Commonwealth of the Northern Mariana Islands. Hawaii J Health Soc Welf 2020, 79, 112-116.

  1. What are the common successful refusing reasons or useful ways for the adolescents to avoid such kind of smoking offers?

Response: Common successful resistance strategies include saying no, leaving the situation, providing an explanation, or avoiding situations where substance use offers are likely to occur.  We have added the following statement with appropriate references to address this (see lines 410-413): “Such strategies, which include refusing or saying no, leaving and avoiding situations where offers are made, or providing an explanation to turn down an offer, have been found to be effective in reducing substance use among youths in both the U.S. and abroad [6, 13, 14, 15].”  

  1. Hecht M.L.; Marsiglia, F. F.; Elek, E.; Wagstaff, D. A.; Kulis, S.; Dustman, P.; Miller-Day, M. Culturally grounded substance use prevention: an evaluation of the keepin' it R.E.A.L. curriculum. Prev Sci, 2003, 40(6), 853-859 doi: 10.1023/a:1026016131401

  1. Kulis, S. S.; Marsiglia, F. F.; Porta, M.; Arévalo Avalos, M. R.; Ayers, S. L. Testing the keepin’ it REAL substance use prevention curriculum among early adolescents in Guatemala City. Prev Sci 2019, 20, 532-543.

  1. Marsiglia, F. F.; Kulis, S.; Booth, J. M.; Nuño Gutiérrez, B.; Robbins, D. E. Long-term effects of the keepin’ it REAL model program in Mexico: Substance use trajectories of Guadalajara middle school students. J Prim Prev 2014, 36, 93-104.
  2. Marsiglia, F. F.; Kulis, S.; Kiehne, E.; Ayers, S. L.; Ricalde, C. A. L.; Sulca, L. B. Adolescent substance-use prevention and legalization of marijuana in Uraguay: A feasibility trial of the keepin’ it REAL prevention program. J Subst Use 2018, 23, 457-465.
  3. Why "training young adolescents from Guam on ways to navigate through substance use offers" other than training adolescents to avoid cigarrettes or e-cigarrettes smoking offer?

Response: Thanks for this suggestion. We have edited that sentence on lines 24-25 as follows (note we feel that “resist” is a more appropriate word to use than “avoid”): “Based on the findings, we provide brief suggestions on developing a school-based prevention curriculum focused on training young adolescents from Guam on ways to resist offers of cigarettes, e-cigarettes, and betel nut.”

  1. In addition, please check the proportion of unavoidable and avoidable IC offers. Are they all equal to 90%?

Response: We believe Reviewer 2 is referring to proportions of the unavoidable and avoidable IC offers reported on Table 1. They are both correctly reported to equal to 90%. In addition, we have checked the proportion of both unavoidable and avoidable IC offers reported both in-text and on all the tables. They are all correct as well. We wish to note that the percentages presented on the tables represent the proportion of focus groups that reported experiencing the particular type of offer—e.g., for the results presented on Table 1, 100% (or all) of the focus groups reported experiencing IC offers, 90% respectively reported experiencing unavoidable IC offers and avoidable IC offers. While one focus group did not report experiencing an unavoidable IC offer and another focus group did not report experiencing an avoidable IC offer, when both unavoidable and avoidable offers are accounted for, all focus groups (100%) had reported experiencing an IC offer.